# FLEXIBLE PRIOR DISTRIBUTIONS FOR DEEP GENERATIVE MODELS

## ABSTRACT

We consider the problem of training generative models with deep neural networks as generators, i.e. to map latent codes to data points. Whereas the dominant paradigm combines simple priors over codes with complex deterministic models, we argue that it might be advantageous to use more flexible code distributions. We demonstrate how these distributions can be induced directly from the data. The benefits include: more powerful generative models, better modeling of latent structure and explicit control of the degree of generalization.

## 1 INTRODUCTION

Generative models have recently moved to the center stage of deep learning in their own right. Most notable is the seminal work on Generative Adversarial Networks (GAN) (Goodfellow et al., 2014) as well as probabilistic architectures known as Variational Autoencoder (VAE) (Kingma & Welling, 2013; Rezende et al., 2014). Here, the focus has moved away from density estimation and towards generative models that – informally speaking – produce samples that are perceptually indistinguishable from samples generated by nature. This is particularly relevant in the context of high-dimensional signals such as images, speech, or text.

Generative models like GANs typically define a generative model via a deterministic generative mechanism or generator $G_\phi : \mathbb{R}^d \rightarrow \mathbb{R}^m$, $\mathbf{z} \mapsto G(\mathbf{z}) = \mathbf{x}$, parametrized by $\phi$. They are often implemented as a deep neural network (DNN), which is hooked up to a code distribution $\mathbf{z} \sim \mathcal{P}_{\mathbf{z}}$, to induce a distribution $\mathbf{x} \sim \mathcal{P}_{\mathbf{x}}$. It is known that under mild regularity conditions, by a suitable choice of generator, any $\mathcal{P}_{\mathbf{x}}$ can be obtained from an arbitrary *fixed* $\mathcal{P}_{\mathbf{z}}$ (Kallenberg, 2006). Relying on the power and flexibility of DNNs, this has led to the view that code distributions should be simple and *a priori* fixed, e.g. $\mathcal{P}_{\mathbf{z}} = \mathcal{N}(\mathbf{0}, \mathbf{I})$. As shown in Arjovsky & Bottou (2017) for DNN generators, $\mathrm{Im}(G_\phi)$ is a countable union of manifolds of dimension $d$ though, which may pose challenges, if $d < m$. Whereas a current line of research addresses this via alternative (non-MLE or KL-based) discrepancy measures between distributions (Dziugaite et al., 2015; Nowozin et al., 2016; Arjovsky et al., 2017), we investigate an orthogonal direction:

**Claim 1.** *It is advantageous to increase the modeling power of a generative model, not only via $G_\phi$, but by using more flexible prior code distributions $\mathcal{P}_{\mathbf{z}}$.*

Another potential benefit of using a flexible latent prior is the ability to reveal richer structure (e.g. multimodality) in the latent space via $\mathcal{P}_{\mathbf{z}}$, a view which is also supported by evidence on using more powerful posterior distributions (Mescheder et al., 2017). This argument can also be understood as follows. Denote by $\mathcal{Q}_{\mathbf{x}}$ the distribution induced by the generator. Our goal is to ensure the $\mathcal{Q}_{\mathbf{x}}$ distribution matches the true data distribution $\mathcal{P}_{\mathbf{x}}$. This brings us to consider the KL-divergence of the joint distributions which can be decomposed as

$$\mathrm{KL}(\mathcal{P}(\mathbf{x}, \mathbf{z}) \| \mathcal{Q}(\mathbf{x}, \mathbf{z})) = \mathrm{KL}(\mathcal{P}(\mathbf{z}) \| \mathcal{Q}(\mathbf{z})) + \mathbb{E}_{\mathbf{z}} \mathrm{KL}(\mathcal{P}(\mathbf{x}|\mathbf{z}) \| \mathcal{Q}(\mathbf{x}|\mathbf{z})). \tag{1}$$

Assuming that the generator is powerful enough to closely approximate the data's generative process, then the contribution of the term $\mathrm{KL}(\mathcal{P}(\mathbf{x}|\mathbf{z}) \| \mathcal{Q}(\mathbf{x}|\mathbf{z}))$ vanishes or becomes extremely small, and what remains is the divergence between the priors. This means that in light of using powerful neural networks to model $\mathcal{Q}(\mathbf{x}|\mathbf{z})$, the prior agreement becomes a way to assess the quality of our learned model.

Empowered by this quantitative metric to evaluate the modeling power of a generative model, we will demonstrate some deficiencies in the assumption of using an arbitrary fixed prior such as a Normal distribution. We will further validate this observation by demonstrating that a flexible prior can be learned from data by mapping back the data points to their latent space representations. This procedure relies on a (trained) generator to compute an approximate inverse map $H : \mathbb{R}^m \to \mathbb{R}^d$ such that $H \circ G_\phi \approx \text{id}$.

**Claim 2.** *The generator $G_\phi$ implicitly defines an approximate inverse, which can be computed with reasonable effort using gradient descent and without the need to co-train a recognition network. We call this approach generator reversal.*

Note that, if the above argument holds, we can easily find latent vectors $\mathbf{z} = H(\mathbf{x})$ corresponding to given observations $\mathbf{x}$. This then induces an empirical distribution of "natural" codes. An extensive set of experiments presented in this paper reveals that this induced prior yields strong evidence of improved generative models. Our findings clearly suggest that further work is needed to develop flexible latent prior distributions in order to achieve generative models with greater modeling power.

## 2 MEASURING THE MODELING POWER OF THE LATENT PRIOR

### 2.1 GRADIENT–BASED REVERSAL

Let us begin with making Claim 2 more precise. Given a data point $\mathbf{x}$, we aim to compute some approximate code $H(\mathbf{x})$ such that $(G \circ H)(\mathbf{x}) = G(H(\mathbf{x})) =: \tilde{\mathbf{x}} \approx \mathbf{x}$. We do so by simple gradient descent, starting from some random initialization for $\mathbf{z}$ (see Algorithm 1).

---
**Algorithm 1** Generator Reversal

**input** Data point $\mathbf{x}$, loss function $\ell$, initial value $\mathbf{z}_0$
 1: Initialize $\mathbf{z} \leftarrow \mathbf{z}_0$
 2: **repeat**
 3:    $\hat{\mathbf{x}} = G_\phi(\mathbf{z})$          {run generator}
 4:    $\mathbf{z} \leftarrow \mathbf{z} - \eta \nabla_\mathbf{z} \ell(\mathbf{z}, \mathbf{x}), \;\; \ell(\mathbf{z}, \mathbf{x}) = \hat{\ell}(\hat{\mathbf{x}}, \mathbf{x})$       {backpropagate error}
 5: **until** converged
**output** latent code $\mathbf{z}$

---

Section B in the Appendix demonstrates that the generator reversal approach presented in Algorithm 1 ensures local convergence of gradient descent for a suitable choice of loss function.

Given the generator reversal procedure presented in Algorithm 1, a key question we would like to address is whether good (low loss) codevectors exist for data points $\mathbf{x}$. First of all, whenever $\mathbf{x}$ was actually generated by $G_\phi$, then surely we know that a perfect, zero-loss pre-image $\mathbf{z}$ exists. Of course finding it exactly would require the exact inverse function of the generator process but our experiments demonstrate that, in practice, an approximate answer is sufficient.

Secondly, if $\mathbf{x}$ is in the training data, then as $G_\phi$ is trained to mimic the true distribution, it would be suboptimal if any such $\mathbf{x}$ would not have a suitable pre-image. We thus conjecture that learning a good generator will also improve the quality of the generator reversal, at least for points $\mathbf{x}$ of interest (generated or data). Note that we do not explicitly train the generator to produce pre-images that would further optimize the training objective. This would require backpropagating through the reversal process which is certainly possible and would likely yield further improvements.

Anecdotally, we have found the generator reversal procedure to be quite effective at finding reasonable pre-images of data samples even for generators initialized for random weights. Some empirical results are provided in Section C of the Appendix.

### 2.2 MEASURING PRIOR AGREEMENT

Modeling the distribution of a complex set of data requires a rich family of distributions which is typically obtained by introducing latent variables $\mathbf{z}$ and modeling the data as $\mathcal{P}(\mathbf{x}) = \int_\mathbf{z} \mathcal{P}(\mathbf{z})\mathcal{P}(\mathbf{x} \mid \mathbf{z})d\mathbf{z}$. Often the prior $\mathcal{P}(\mathbf{z})$ is assumed to be a Normal distribution, i.e. $\mathcal{P}(\mathbf{z}) \sim \mathcal{N}(0, \sigma^2 \mathbf{I})$.

This principle underlies most modern generative models such as GANs, effectively turning the generation process as mapping latent noise $\mathbf{z}$ to observed data $\mathbf{x}$ through a generative model $\mathcal{P}(\mathbf{x} \mid \mathbf{z})$. In GANs, the generative model - or *generator* - is a neural network parameterized as $\mathcal{P}_\theta(\mathbf{x} \mid \mathbf{z})$, and optimized to find the best set of parameters $\theta^*$. Note that an implicit assumption that is made by this modeling choice is that the generative process is adequate and therefore sufficiently powerful to find $\theta^*$ in order to reconstruct the data $\mathbf{x}$. For GANs to work, this assumption has to hold despite the fact that the prior is kept fixed. In theory, such generator does exist as neural networks are universal approximators and can therefore approximate any function. However, finding $\theta^*$ requires solving a high-dimensional and non-convex optimization problem and is therefore not guaranteed to succeed.

We here explore an orthogonal direction. We assume that we have found a suitable generative model $\mathcal{P}_{\theta^*}(\mathbf{x} \mid \mathbf{z})$ that could produce the data distribution $\mathcal{P}(\mathbf{x})$ but the prior is not appropriate. We would like to quantify to what degree does the assumed prior $\mathcal{P}(\mathbf{z}) \sim \mathcal{N}(0, \sigma^2 \mathbf{I})$ disagrees with the *data induced* prior therefore measuring how well our generated distribution agrees with the data distribution. Our goal in doing so is not to propose a new training criterion, but rather to assess the quality of our generative model by measuring the quality of the prior.

**Prior Agreement (PAG) Score**  We consider the standard case where $\mathcal{P}(\mathbf{z})$ is modeled as a multivariate Normal distribution $\mathcal{N}(0, \sigma^2 \mathbf{I})$ with diagonal uniform covariance. Our goal is to measure the disagreement between this prior and some more suitable prior. The latter not being known a-priori, we instead settle for a multivariate Normal with diagonal covariance $\mathcal{N}(0, \boldsymbol{\Sigma}), \boldsymbol{\Sigma} := \text{diag}(\nu_i^2)$ where the $\nu_i$ are inferred from a trained generator as detailed below. This choice of prior will allow for a simple computation of divergence as follows:

$$
\begin{aligned}
\text{KL}(\mathcal{N}(0, \boldsymbol{\Sigma}) \| \mathcal{N}(0, \sigma^2 \mathbf{I})) &= \frac{1}{2} \left( \frac{1}{\sigma^2} \text{tr}(\Sigma) - d - (\log |\Sigma| - \log \sigma^{2d}) \right) \\
&= \frac{1}{2} \left( \frac{1}{\sigma^2} \text{tr}(\Sigma) - d - \log \prod_i^d \frac{\nu_i^2}{\sigma^2} \right) \\
&= \frac{1}{2} \left( \sum_i^d \frac{\nu_i^2}{\sigma^2} - d - \sum_i^d \log \frac{\nu_i^2}{\sigma^2} \right) \\
&= \frac{1}{2} \sum_i^d \left[ \frac{\nu_i^2}{\sigma^2} - \log \frac{\nu_i^2}{\sigma^2} - 1 \right]
\end{aligned}
\tag{2}
$$

The divergence defined in Equation 2 defines the *Prior Agreement (PAG) Score*. It requires the quantities $\nu_i^2$ which can easily be computed by mapping the data to the latent space using the reversal procedure described in Section 2.1 and then performing an SVD decomposition on the resulting latent vectors $\hat{\mathbf{z}} = \text{Generator Reversal}(\mathbf{x})$. The $\nu_i$ then correspond to the singular values $diag(\Sigma)$ obtained in the SVD decomposition $\hat{\mathbf{z}} = \mathbf{U}\Sigma\mathbf{V}^*$.

Note that more complex choices as a substitute for the data induced prior would allow for a better characterization of the inadequacy of the Normal prior with uniform covariance. We will however demonstrate that the PAG score defined in Equation 2 is already effective at revealing surprising deficiencies in the choice of the Normal prior.

## 3  LEARNING THE DATA INDUCED PRIOR

So far, we have introduced a way to characterize the fit of a chosen prior $\mathcal{P}(\mathbf{z})$ to model the data distribution $\mathcal{P}(\mathbf{x})$ given a trained generator $\mathcal{P}(\mathbf{x} \mid \mathbf{z})$. Equipped with this agreement score, we now turn our attention to designing a method to address the potential problems that could arise from choosing an inappropriate prior $\mathcal{P}(\mathbf{z})$. As shown in Figure 1, we suggest learning the data induced prior distribution using a secondary GAN we name PGAN which learns a mapping function $h : \mathcal{Z}' \to \mathcal{Z}$ where $\mathcal{Z}' \subseteq \mathbb{R}^{d'}$ is an auxiliary latent space with the same or higher ambient dimension as the original latent space $\mathcal{Z} \subseteq \mathbb{R}^d$. The mapping $h$ defines a transformation of the noise vectors in order to match the data induced prior. Note that training the mapping function $h$ is done by keeping the original GAN unchanged, thus we only need to run the reversal process once for the dataset and then the reverted data in the latent space becomes the target to train $h$.

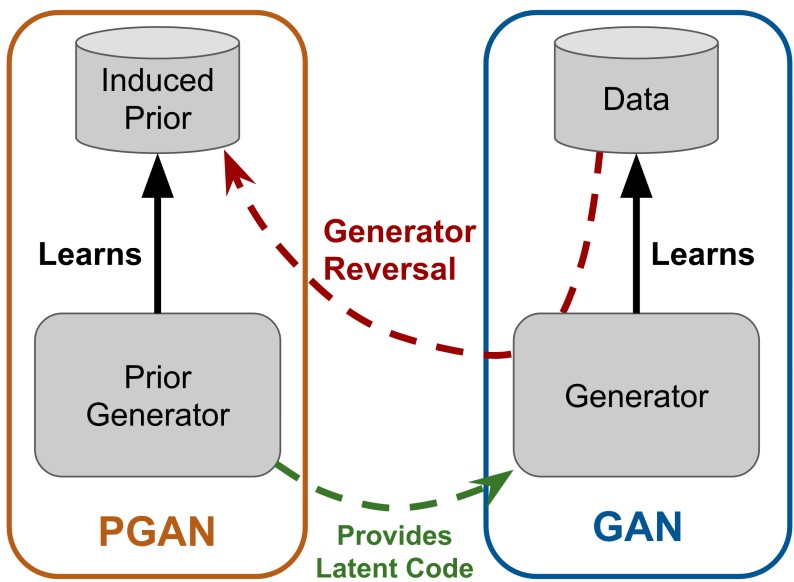

Figure 1: A data induced prior distribution is learned using a secondary GAN named PGAN. This prior is then used to further train the original GAN.

If the original prior over the latent space $\mathcal{Z}$ is indeed not a good choice to model $\mathcal{P}(\mathbf{x})$, we should see evidence of a better generative model by sampling from the transformed latent space $\mathcal{Z}'$. Such evidence including better quality samples, less outliers and higher PAG scores are shown in Figure 5 (see details in Section 4). Note that having obtained this mapping opens the door to various schemes such as multiple rounds of re-training the original GAN and training the PGAN using the newly learned prior or training yet another PGAN to match the data induced prior of the first PGAN. We leave these practical considerations to future work, as our goal is simply to provide a method to quantify and remedy the fundamental problem of prior disagreement.

## 4 EXPERIMENTS

The experimental results presented in this section are based on off-the-shelf GAN architectures whose details are provided in the appendix. We restrict the dimension of the latent space to $d = 20$. Although similar experimental results can be obtained with latent spaces of larger dimensions, the low-dimensional setting is particularly interesting as it requires more compression, providing an ideal scenario to empirically verify the fitness of the fixed Gaussian prior with respect to the data induced prior.

### 4.1 MAPPING A DATASET TO THE LATENT SPACE

Given a generator network, we first map 1024 data points from the MNIST dataset to the latent space using the Generator Reversal procedure. We then use t-SNE to reduce the dimensionality of the latent vectors to 2 dimensions. We perform this procedure for both an untrained and a fully-trained networks and show the results in Figure 2. One can clearly see a multi-modal structure emerging after training the generator network, indicating that a unimodal Normal distribution is not an appropriate choice as a prior over the latent space.

### 4.2 PRIOR-DATA-DISAGREEMENT SAMPLES

In order to demonstrate that a simple Normal prior does not capture the data induced prior, we sample points that are likely under the Normal prior but unlikely under the data induced prior. This is achieved by sampling a batch of 1000 samples from the Normal prior, then ordering the samples according to their mean squared distance to the found latent representations of a batch of data.

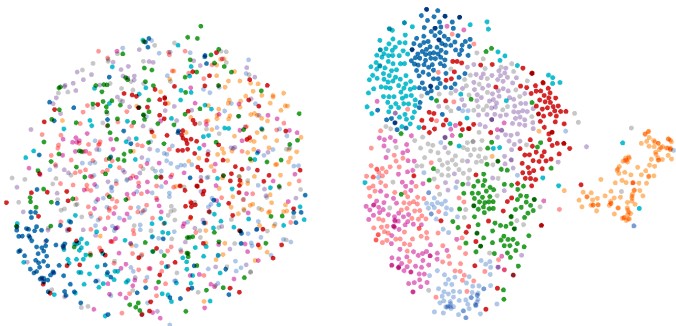

Figure 2: *Generator reversal on a sample of 1024 MNIST digits. Projections of data points with an untrained (left) and a fully trained GAN (right). Colors represent the respective class labels. The ratios of between-cluster distances to within-cluster distances are 0.1 (left) and 1.9 (right).*

Figure 3 shows the the top 20 samples in the data space (obtained after mapping the top 20 latent vectors to the data space using the generator). The poor quality of these samples indicate that the induced prior assigns loss probability mass to unlikely samples.

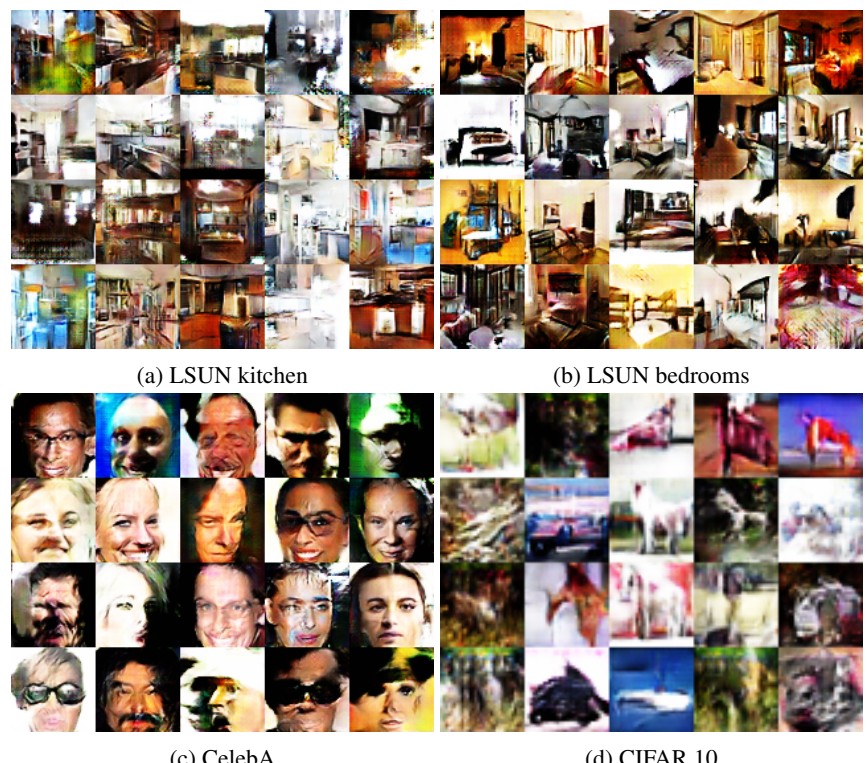

(a) LSUN kitchen

(b) LSUN bedrooms

(c) CelebA

(d) CIFAR 10

Figure 3: Prior-data-disagreement samples. We visualize samples for which the likelihood under the GAN prior is high, but low under the data-induced prior. Note that most of these samples are of poor visual quality and contain numerous artifacts.

## 4.3 EVALUATING THE PAG SCORE

We now evaluate how the PAG score correlates with the visual quality of the samples produced by a generator. We first train a selection of 16 GAN models using different combinations of filter size, layer size and regularization constants. We then select the best and the worst model by visually

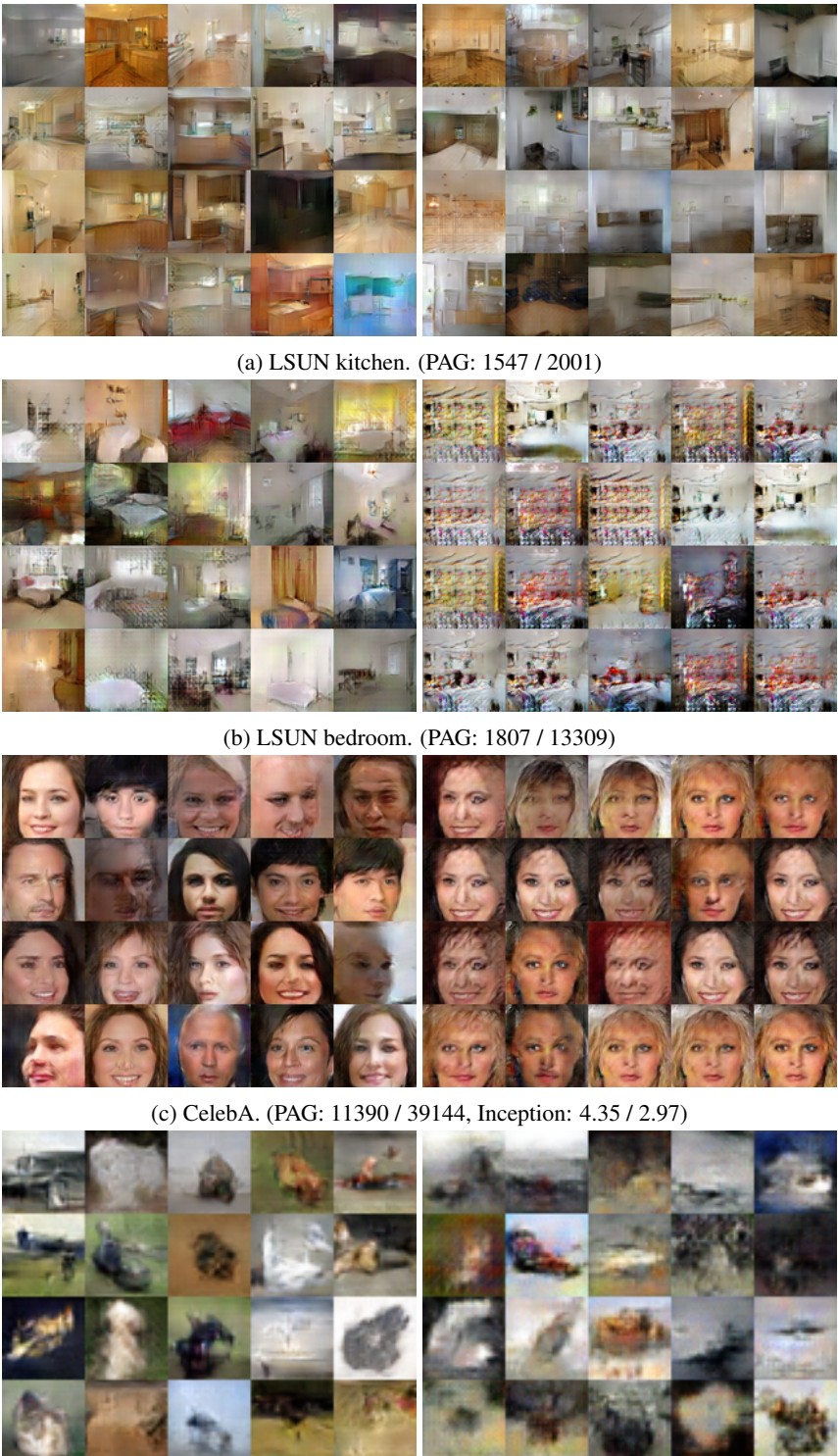

(a) LSUN kitchen. (PAG: 1547 / 2001)

(b) LSUN bedroom. (PAG: 1807 / 13309)

(c) CelebA. (PAG: 11390 / 39144, Inception: 4.35 / 2.97)

(d) CIFAR 10. (PAG: 5021 / 5352, Inception: 6.30 / 5.94)

Figure 4: Best / worst selection of samples (by visual inspection) from a number of different GAN models. We also report the PAG and Inception Scores (when available). Note that the PAG score agrees with the Inception Score, but does not require labeled data to be evaluated.

inspecting the generated samples. We show the samples as well as the corresponding PAG scores in Figure 4. These results clearly demonstrate that the PAG score strongly correlates with the visual quality of the samples. We also report the Inception score for datasets that provide a class label, and observe a strong agreement with the PAG scores.

## 4.4 Learning the Data Induced Prior using a secondary GAN

Following the procedure presented in Section 3, we train a GAN until convergence and then use the Generator Reversal procedure to map the dataset to the latent space, therefore inducing a *data-induced prior*. We then train a secondary GAN (called PGAN) to learn this prior from which we can then continue training the original GAN for a few steps.

We expect that the model trained with the data-induced prior will be better at capturing the true data distribution. This is empirically verified in Figure 5 by inspecting samples produced by the original and re-trained model. We also report the PAG scores for which we see a significant reduction therefore confirming our hypothesis of obtaining an improved generative model. Note that the data-induced prior yields more realistic and varied output samples, even though it uses the same dimensionality of latent space as the original simple prior.

## 5 Related Work

Our generator reversal is similar in spirit to Kindermann & Linden (1990), but their intent differs as they use this technique as a tool to visualize the information processing capability of a neural network. Unlike previous works that require the transfer function to be bijective Baird et al. (2005); Rippel & Adams (2013), our approach does not strictly have this requirement, although this could still be imposed by carefully selecting the architecture of the network as shown in Dinh et al. (2016); Arjovsky et al. (2017).

In the context of GANs, other works have used a similar reversal mechanism as the one used in our approach, including e.g. Che et al. (2016); Dumoulin et al. (2016); Donahue et al. (2016). All these methods focus on training a separate *encoder* network in order to map a sample from the data space to its latent representation. Our goal is however different as the reversal procedure is here used to estimate a more flexible prior over the latent space.

Finally, we note that the importance of using an appropriate prior for GAN models has also been discussed in Han et al. (2016) which suggested to infer the continuous latent factors in order to maximize the data log-likelihood. However this approach still makes use of a simple fixed prior distribution over the latent factors and does not use the inferred latent factors to construct a prior as suggested by our approach.

## 6 Conclusion

We started our discussion by arguing that is advantageous to increase the modeling power of a generative model by using more flexible prior code distributions. We substantiated our claim by deriving a quantitative metric estimating the modeling power of a fixed prior such as the Normal prior commonly used when training GAN models. Our experimental results confirm that this measure reveals the standard choice of an arbitrary fixed prior is not always an appropriate choice. In order to address this problem, we presented a novel approach to estimate a flexible prior over the latent codes given by a generator $G_\phi$. This was achieved through a reversal technique that reconstruct latent representations of data samples and use these reconstructions to construct a prior over the latent codes. We empirically demonstrated that the resulting data-induced prior yields several benefits including: more powerful generative models, better modeling of latent structure and semantically more appropriate output.

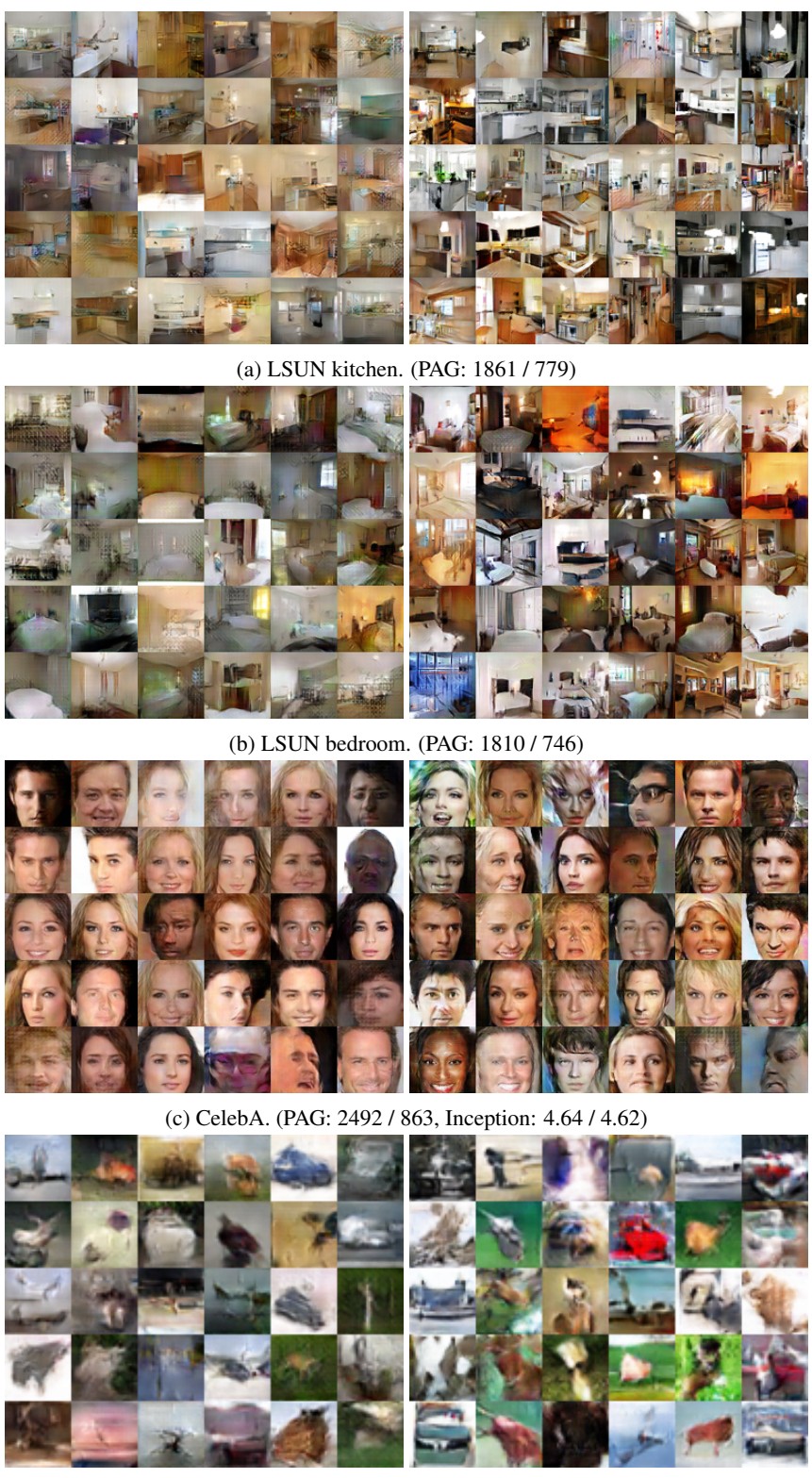

(a) LSUN kitchen. (PAG: 1861 / 779)

(b) LSUN bedroom. (PAG: 1810 / 746)

(c) CelebA. (PAG: 2492 / 863, Inception: 4.64 / 4.62)

(d) CIFAR 10. (PAG: 2485 / 1064, Inception: 6.24 / 6.59)

Figure 5: Samples before (left) and after (right) training with the data induced prior. Note the increased level of diversity in the samples obtained from the induced prior.

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

# APPENDIX

## A  DERIVATION EQUATION 1

$$
\begin{aligned}
\mathrm{KL}(p(\mathbf{x}, \mathbf{z}) \| q(\mathbf{x}, \mathbf{z})) &= \mathbb{E}_{\mathbf{x}, \mathbf{z}} \log \frac{p(\mathbf{x}, \mathbf{z})}{q(\mathbf{x}, \mathbf{z})} \\
&= \mathbb{E}_{\mathbf{x}, \mathbf{z}} \log \frac{p(\mathbf{x}|\mathbf{z}) p(\mathbf{z})}{q(\mathbf{x}|\mathbf{z}) q(\mathbf{z})} \\
&= \mathbb{E}_{\mathbf{x}, \mathbf{z}} \log \frac{p(\mathbf{z})}{q(\mathbf{z})} + \mathbb{E}_{\mathbf{x}, \mathbf{z}} \log \frac{p(\mathbf{x}|\mathbf{z})}{q(\mathbf{x}|\mathbf{z})} \\
&= \mathbb{E}_{\mathbf{z}} \log \frac{p(\mathbf{z})}{q(\mathbf{z})} + \mathbb{E}_{\mathbf{z}} \mathbb{E}_{\mathbf{x}|\mathbf{z}} \log \frac{p(\mathbf{x}|\mathbf{z})}{q(\mathbf{x}|\mathbf{z})} \\
&= \mathrm{KL}(p(\mathbf{z}) \| q(\mathbf{z})) + \mathrm{KL}(p(\mathbf{x}|\mathbf{z}) \| q(\mathbf{x}|\mathbf{z}))
\end{aligned}
\tag{3}
$$

## B  LOCAL CONVERGENCE OF THE GRADIENT–BASED REVERSAL

Let us demonstrate that the generator reversal approach presented in Algorithm 1 ensures local convergence of gradient descent for a suitable choice of loss function.

**Proposition 1.** *We are given an $\ell_2$ loss function $\ell : \mathcal{Z} \times \mathcal{X} \to \mathbb{R}$ and a generator function $G : \mathcal{Z} \to \mathcal{X}$. Consider a point $\mathbf{x}_* = G(\mathbf{z}_*)$ and assume the function $G(\mathbf{z})$ is locally invertible around $\mathbf{z}_*$ [1]. Then the reconstruction problem $\min_{\mathbf{z}} \ell(\mathbf{z}, \mathbf{x})$ is locally convex at $\mathbf{x}_*$.*

*Proof.* We prove the result stated above by showing that the Hessian of $\ell$ at $\mathbf{z}_*$ is positive semidefinite.

$$
\ell(\mathbf{z}, \mathbf{x}_*) = \frac{1}{2} \|G(\mathbf{z}) - \mathbf{x}_*\|^2 = \frac{1}{2} \|G(\mathbf{z}) - G(\mathbf{z}_*)\|^2
\tag{4}
$$

Let $J_G(\mathbf{z})$ denote the Jacobian of $G(\mathbf{z})$ and let's compute the Hessian of $\ell$ at $\mathbf{z}_*$:

$$
\begin{aligned}
\nabla_{\mathbf{z}} \ell(\mathbf{z}, \mathbf{x}_*) &= \mathbf{J}_G(\mathbf{z}) \left(G(\mathbf{z}) - G(\mathbf{z}_*)\right) \\
\implies \nabla_{\mathbf{z}}^2 \ell(\mathbf{z}, \mathbf{x}_*) &= \nabla_{\mathbf{z}}^2 G(\mathbf{z}) \left(G(\mathbf{z}) - G(\mathbf{z}_*)\right) + \mathbf{J}_G(\mathbf{z}) \mathbf{J}_G(\mathbf{z})^\top \\
\implies \nabla_{\mathbf{z}}^2 \ell(\mathbf{z}_*, \mathbf{x}_*) &= \mathbf{0} + \mathbf{J}_G(\mathbf{z}_*) \mathbf{J}_G(\mathbf{z}_*)^\top
\end{aligned}
$$

Since $G(\mathbf{z})$ is assumed to be locally invertible around $\mathbf{z}^*$, then $\mathbf{J}_G(\mathbf{z}^*) \neq \mathbf{0}$ and the Hessian $\nabla_{\mathbf{z}}^2 \ell(\mathbf{z}_*)$ is therefore positive semidefinite. $\qquad\square$

Note that one could also add an $\ell_2$ regularizer to Equation 4 in order to obtain a locally strongly-convex function.

## C  RANDOM NETWORK EXPERIMENTS

It is very hard to give quality guarantees for the approximations obtained via generator reversal. Here, we provide experimental evidence by showing that even a DNN generator with random weights $\phi$ can provide reasonable pre-images for data samples. As we argued above, we believe that actual training of $G_\phi$ will improve the quality of pre-images, so this is in a way a worst case scenario.

Examples for three different image data sets are shown in Figure 6. Here we show the average reconstruction error as a function of the number of gradient update steps. We observe that the error decreases steadily as the reconstruction progresses and reaches a low value very quickly. We also show randomly selected reconstructed samples in Figure 7, which reflect the fast decrease in

---

[1] Note that this is a less restrictive assumption than the diffeomorphism property required in Arjovsky & Bottou (2017)

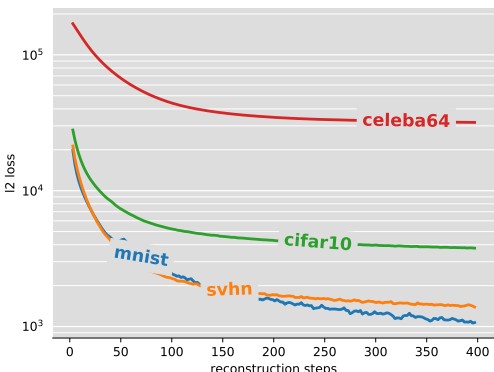

**Reconstruction loss in random networks**

Figure 6: *Reconstruction loss in generator networks with random weights.*

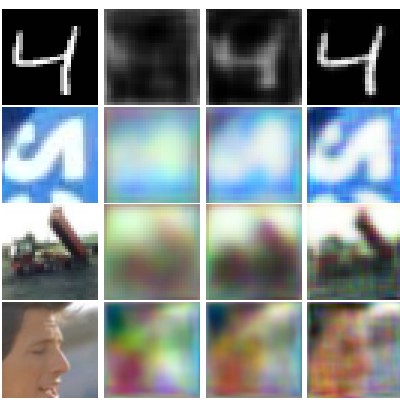

Figure 7: *Reconstruction quality using generator networks with random weights. The left column is the original image, followed by reconstructions after 5, 20 and 400 steps.*

terms of reconstruction error. After only 5 update steps, one can already recognize the general outline of the picture. This is perhaps even more surprising considering that these results were obtained using a generator with *completely random weights*. A similar finding was also reported in He et al. (2016) which constructed deep visualizations using untrained neural networks initialized with random weights.

## D    LATENT SPACE DATA DISTRIBUTION

Figure 8 shows the distribution of the obtained latent codes after Generator Reversal. Interestingly, although the distributions are different from the naive prior, they are not indicative of low rank latent data. This agrees with our expectations, as a well trained generator will make use of all available latent dimensions.

## E    DETAILED EXPERIMENT SETUP

Our experimental setup closely follows popular setups in GAN research in order to facilitate reproducibility and enable qualitative comparisons of results. Our network architectures are as follows:

The generator samples from a latent space of dimension 20, which is fed through a padded deconvolution to form an initial intermediate representation of $4 \times 4 \times 512$, which is then fed through four layers of deconvolutions with 512, 256, 128 and 64 filters, followed by a last deconvolution to get to the desired output size and channels.

The discriminator consists of three layers of convolutional layers with 512, 256, 128 and 64 filters, followed by a fully connected layer and a sigmoid classifier.

Both the generator and the discriminator use $4 \times 4$ filters with a stride of $2$ in order to up- and downscale the representations, respectively. The generator employs ReLU non-linearities, except for the last layer, which uses hyperbolic tangent. The discriminator uses Leaky ReLU non-linearities with a leak of $0.2$, which is standard in the GAN literature.

The PGAN consists of four layers of fully connected units in both the generator and discriminator. Apart from the layers being fully connected, the architecture is analogous to the original GAN.

We use RMSProp(Tieleman & Hinton, 2012) with a step size of $0.0003$ and mini-batches of size 100 for optimization for all networks.

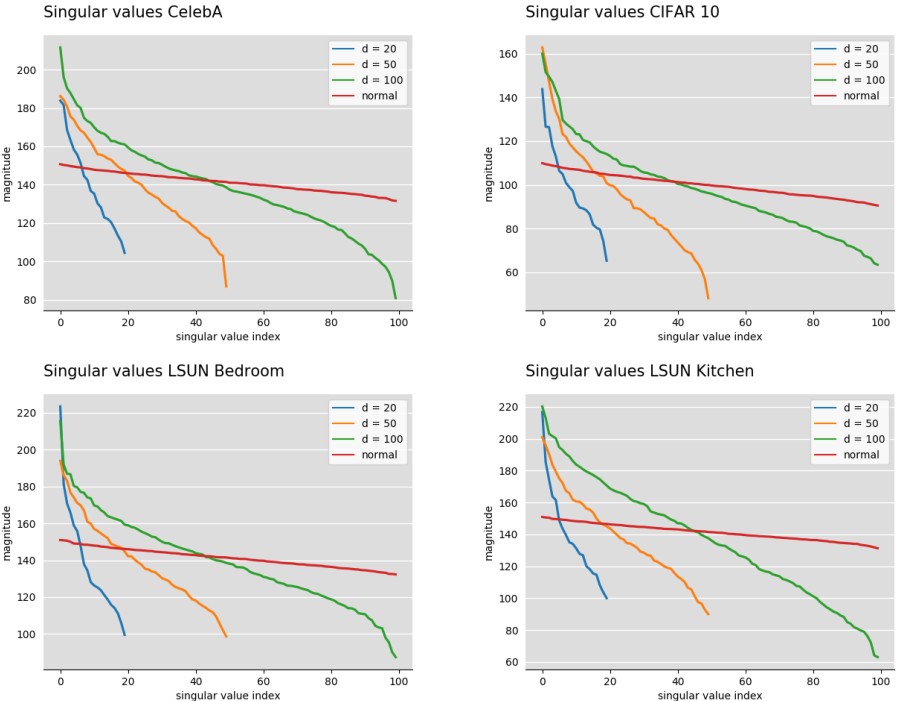

Figure 8: *Distribution of singular values (in GANs using d latent dimensions) used in calculation of the PAG scores. For comparison, singular values of a sample of normally distributed latent codes in 100 dimensions are shown.*

For the generator reversal process, we use a learning rate of $0.05$. The initial noise vectors are sampled from a normal distribution with $\sigma = 0.0001$.

We train until we can no longer see any significant qualitative improvement in the generated images or any quantitative improvement in the inception score (if available).

For the CelebA dataset, we crop the images to a size of $118 \times 118$ pixels, after which we resize them to $64 \times 64$ pixels.

