# OpenReview forum: "Flexible Prior Distributions for Deep Generative Models"
_ICLR.cc/2018/Conference — Reject_

### Official Review · AnonReviewer2 · 2017-11-27
**review for flexible priors for GAN**

**Rating:** 6
**Confidence:** 4

**Review:**

Summary:

The paper proposes to learn new priors for latent codes z  for GAN training.  for this the paper shows that there is a mismatch between the gaussian prior and an estimated of the latent codes of real data by reversal of the generator . To fix this the paper proposes to learn a second GAN to learn the prior distributions of "real latent code" of the first GAN. The first GAN then uses the second GAN as prior to generate the z codes.

Quality/clarity:

The paper is well written and easy to follow.

Originality:

pros:
-The paper while simple sheds some light on important problem with the prior distribution used in GAN.
- the second GAN solution trained on reverse codes from real data is interesting
- In general the topic is interesting, the solution presented is simple but needs more study

cons:

- It related to adversarial learned inference and BiGAN, in term of learning the mapping  z ->x, x->z and seeking the agreement.
- The solution presented is not end to end (learning a prior generator on learned models have been done in many previous works on encoder/decoder)

General Review:

More experimentation with the latent codes will be interesting:

- Have you looked at the decay of the singular values of the latent codes obtained from reversing the generator? Is this data low rank? how does this change depending on the dimensionality of the latent codes? Maybe adding plots to the paper can help.

- the prior agreement score is interesting but assuming gaussian prior also for the learned latent codes from real data is maybe not adequate.  Maybe computing the entropy of the codes using a nearest neighbor estimate of the entropy  can help understanding the entropy difference wrt to the isotropic gaussian prior?

- Have you tried to multiply the isotropic normal noise with the learned singular values and generate images from  this new prior  and compute inceptions scores etc? Maybe also rotating the codes with the singular vector matrix V or \Sigma^{0.5} V?

- What architecture did you use for the prior generator GAN?

- Have you thought of an end to end way to learn the prior generator GAN?

****** I read the authors reply. Thank you for your answers and for the SVD plots this is  helpful.  *****

---

> ### Author Response · Authors · 2018-01-05
> **Thank you**
>
> Thank you for the comments. We invite you to have a look at our appendix, which now includes experiments you suggested.
>
> - It related to adversarial learned inference and BiGAN, in term of learning the mapping  z ->x, x->z and seeking the agreement.
>
> We agree that there is a relation, but also there are fundamental differences in our motivation and the approach itself. Most importantly, we do not learn the mapping x -> z, but we instead rely on a deterministic procedure for doing so.
>
>
> - Have you looked at the decay of the singular values of the latent codes obtained from reversing the generator? Is this data low rank? how does this change depending on the dimensionality of the latent codes? Maybe adding plots to the paper can help.
>
> We have updated the paper to include plots of the distribution of singular values in different dimensional latent spaces (see appendix, figure 8). It appears that the reconstructed latent codes are not low rank, agreeing with what one would expect from a well-trained generator.
>
>
> - the prior agreement score is interesting but assuming gaussian prior also for the learned latent codes from real data is maybe not adequate.  Maybe computing the entropy of the codes using a nearest neighbor estimate of the entropy  can help understanding the entropy difference wrt to the isotropic gaussian prior?
>
> We have experimented with nearest neighbor methods, but found them unreliable for high-dimensional spaces. Note that the reason we use a diagonal gaussian for the PAG scores is not that we propose this to be the best prior, but because it is a single step in complexity above the naive prior. If we find a discrepancy between these two, than we also know that the naive prior is inferior to any even more complex prior.
>
>
> - Have you tried to multiply the isotropic normal noise with the learned singular values and generate images from  this new prior  and compute inceptions scores etc? Maybe also rotating the codes with the singular vector matrix V or \Sigma^{0.5} V?
>
> As mentioned, our intention is not to use the non-isotropic gaussian as a prior in practice, but we have indeed tried this and have not found a significant improvement in either inception scores or visual results.
>
>
> - What architecture did you use for the prior generator GAN?
>
> We briefly describe this in the appendix to be four fully connected layers. We’ve updated the section to clarify that the rest of the architecture (nonlinearities, batch norm, etc.) matches the original GAN.
>
>
> - Have you thought of an end to end way to learn the prior generator GAN?
> It is certainly possible to learn the data induced prior continuously along with the training procedure and we have had good results when trying this ourselves. However, this requires running the reversal procedure in a continuous fashion, rather than just once, and introduces an impractical overhead. Further, we regard such a procedure as a separate contribution from this paper.

---

### Official Review · AnonReviewer3 · 2017-11-27
**Using flexible priors for generative models**

**Rating:** 6
**Confidence:** 3

**Review:**

The paper demonstrates the need and usage for flexible priors in the latent space alongside current priors used for the generator network. These priors are indirectly induced from the data - the example discussed is via an empirical diagonal covariance assumption for a multivariate Gaussian. The experimental results show the benefits of this approach.
The paper provides for a good read.

Comments:

1. How do the PAG scores differ when using a full covariance structure? Diagonal covariances are still very restrictive.
2. The results are depicted with a latent space of 20 dimensions. It will be informative to see how the model holds in high-dimensional settings. And when data can be sparse.
3. You could consider giving the Discriminator, real data etc in Fig 1 for completeness as a graphical summary.

---

> ### Author Response · Authors · 2018-01-05
> **Thank you**
>
> Thanks for the comments. Please see our responses below.
>
> 1. How do the PAG scores differ when using a full covariance structure? Diagonal covariances are still very restrictive.
>
> We have attempted to use full covariances, but more often than not, we ran into numerical issues that made the resulting scores unusable. Note that the use of diagonal covariances for calculating the scores is purposefully chosen to be just a single step in complexity above the naive prior.
>
>
> 2. The results are depicted with a latent space of 20 dimensions. It will be informative to see how the model holds in high-dimensional settings. And when data can be sparse.
>
> The improvement gained from using PGAN slightly decreases in higher dimensions (we have tried up to 200) in terms of visual results, simply because the data induced prior becomes less complex in higher dimensions. However, a discrepancy between the naive prior and the data induced prior remains and is equally measurable.
>
>
> 3. You could consider giving the Discriminator, real data etc in Fig 1 for completeness as a graphical summary.
>
> We originally designed the Figure as you suggested but found the graphic to be too cluttered. Since we assume basic familiarity with GANs throughout the text, we therefore decided to use the “simplified” version provided in our submission.

---

### Official Review · AnonReviewer1 · 2017-12-02
**An interesting idea with a somewhat questionable execution**

**Rating:** 5
**Confidence:** 4

**Review:**

The paper proposes, under the GAN setting, mapping real data points back to the latent space via the "generator reversal" procedure on a sample-by-sample basis (hence without the need of a shared recognition network) and then using this induced empirical distribution as the "ideal" prior targeting which yet another GAN network might be trained to produce a better prior for the original GAN.

I find this idea potentially interesting but am more concerned with the poorly explained motivation as well as some technical issues in how this idea is implemented, as detailed below.

1. Actually I find the entire notion of an "ideal" prior under the GAN setting a bit strange. To start with, GAN is already training the generator G to match the induced P_G(x) (from P(z)) with P_d(x), and hence by definition, under the generator G, there should be no better prior than P(z) itself (because any change of P(z) would then induce a different P_G(x) and hence only move away from the learning target).

I get it that maybe under different P(z) the difficulty of learning a good generator G can be different, and therefore one may wish to iterate between updating G (under the current P(z)) and updating P(z) (under the current G), and hopefully this process might converge to a better solution. But I feel this sounds like a new angle and not the one that is adopted by the authors in this paper.

2. I think the discussions around Eq. (1) are not well grounded. Just as you said right before presenting Eq. (1), typically the goal of learning a DGM is just to match Q_x with the true data distrubution P_x. It is **not** however to match Q(x,z) with P(x,z). And btw, don't you need to put E_z[ ... ] around the 2nd term on the r.h.s. ?

3. I find the paper mingles notions from GAN and VAE sometimes and misrepresents some of the key differences between the two.

E.g. in the beginning of the 2nd paragraph in Introduction, the authors write "Generative models like GANs, VAEs and others typically define a generative model via a deterministic generative mechanism or generator ...". While I think the use of a **deterministic** generator is probably one of the unique features of GAN, and that is certainly not the case with VAE, where typically people still need to specify an explicit probabilistic generative model.

And for this same reason, I find the multiple references of "a generative model P(x|z)" in this paper inaccurate and a bit misleading.

4. I'm not sure whether it makes good sense to apply an SVD decomposition to the \hat{z} vectors. It seems to me the variances \nu^2_i shall be directly estimated from \hat{z} as is. Otherwise, the reference "ideal" distribution would be modeling a **rotated** version of the \hat{z} samples, which imo only introduces unnecessary discrepancies.

5. I don't quite agree with the asserted "multi-modal structure" in Figure 2. Let's assume a 2d latent space, where each quadrant represents one MNIST digit (e.g. 1,2,3,4). You may observe a similar structure in this latent space yet still learn a good generator under even a standard 2d Gaussian prior. I guess my point is, a seemingly well-partitioned latent space doesn't bear an obvious correlation with a multi-modal distribution in it.

6. The generator reversal procedure needs to be carried out once for each data point separately, and also when the generator has been updated, which seems to be introducing a potentially significant bottleneck into the training process.

---

> ### Author Response · Authors · 2018-01-05
> **Very helpful feedback**
>
> Thank you for the detailed feedback. We have made changes to the writeup and would like to address your comments below:
>
> 1. Notion of “Ideal” prior:
>
> We do agree that using the terminology “ideal prior” to refer to the data induced prior might cause confusions and we have now adjusted the writeup accordingly.
> However, we disagree with the statement “there should be no better prior than P(z) itself” where P(z) refers to what we call “naive” prior. The reason is that the generator does not have infinite capacity to map any distribution to any other distribution, but is restricted by its architecture and by the training procedure. We highlighted the resulting discrepancy in our experiments by showing that there exist “empty” regions under the naive prior (figure 3).
> For a perfect generator, moving away from the naive prior would indeed move the generated data away from the learning target, but in practice, we have shown that replacing the naive prior with the data induced prior can actually improve the results significantly (figure 5).
>
>
> 1.5 “one may wish to iterate between updating G (under the current P(z)) and updating P(z) (under the current G), and hopefully this process might converge to a better solution.”
>
> This is indeed a valid procedure and we have done this successfully, but we would like to keep the contribution of this paper focused to justifying a single step in this procedure and therefore did not include these results.
>
>
> 2. I think the discussions around Eq. (1) are not well grounded.
>
> We implicitly argue that matching the joint distributions relates to matching the marginals.
> Indeed, the KL divergence between the joint distributions is trivially a lower bound on the KL divergence between the marginals and since training the generator to convergence will minimize the conditional KL, further improvement can only be made by matching the priors.
>
> 2.5 don't you need to put E_z[ ... ] around the 2nd term on the r.h.s. ?
>
> Absolutely. We have updated the writeup.
>
>
> 3. the paper mingles notions from GAN and VAE sometimes
>
> We have updated the writeup to focus our discussion on GANs (expect in the first paragraph).
>
>
> 4. I'm not sure whether it makes good sense to apply an SVD decomposition to the \hat{z} vectors. It seems to me the variances \nu^2_i shall be directly estimated from \hat{z} as is. Otherwise, the reference "ideal" distribution would be modeling a **rotated** version of the \hat{z} samples, which imo only introduces unnecessary discrepancies.
>
> The SVD is only used to compute the prior agreement score and the use of it is resulting from the definition of the KL between multivariate normals. When we learn the data induced prior, our targets are the reconstructed latent codes as is.
>
>
> 5. I don't quite agree with the asserted "multi-modal structure" in Figure 2. Let's assume a 2d latent space, where each quadrant represents one MNIST digit (e.g. 1,2,3,4). You may observe a similar structure in this latent space yet still learn a good generator under even a standard 2d Gaussian prior. I guess my point is, a seemingly well-partitioned latent space doesn't bear an obvious correlation with a multi-modal distribution in it.
>
> We agree with your statement, but Figure 2 shows a latent space that is not only well-partitioned, but also has empty regions that shouldn’t be empty under the original prior. If there are regions in the latent space that are never used when explicitly reconstructing the data manifold, but the generator samples from all regions equally when learning to match the same data manifold, there must be a multi-modal structure that disagrees with the given prior.
>
>
> 6. The generator reversal procedure needs to be carried out once for each data point separately, and also when the generator has been updated, which seems to be introducing a potentially significant bottleneck into the training process.
>
> The reversal procedure is carried out once per data point indeed, but this only happens once, after the generator has finished training using the naive prior. In addition, this can be carried out using very large batches of data (since no learning takes place during reversal). Thus, the overhead essentially amounts to one large-batch pass over the data in the entire duration of learning.

---

### Decision · Program_Chairs · 2018-01-29
**ICLR 2018 Conference Acceptance Decision**

**Decision:**

Reject

**Comment:**

This paper presents a method for learning more flexible prior distributions for GANs by learning another distribution on top of the latent codes for training examples. It's reminiscent of layerwise training of deep generative models. This seems like a reasonable thing to do, but it's probably not a substantial enough contribution given that similar things have been done for various other generative models. Experiments show improvement in samples compared with a regular GAN, but don't compare against various other techniques that have been proposed for fixing mode dropping. For these reasons, as well as various issues pointed out by the reviewers, I don't recommend acceptance.